# PROCEEDINGS A

complexity, mathematical modelling, applied mathematics

coevolving networks, livestock trade, games, dynamics on networks, dynamics of networks

**Author for correspondence:**
Ewan Colman
e-mail: ecolman@ed.ac.uk

One contribution to a special feature 'A generation of network science' organized by Danica Vukadinovic-Greetham and Kristina Lerman.

# Spontaneous divergence of disease status in an economic epidemiological game

Ewan Colman[1], Nick Hanley[2] and Rowland R. Kao[1]

[1]Royal (Dick) School of Veterinary Studies and the Roslin Institute, University of Edinburgh, Easter Bush, Midlothian, UK
[2]Institute of Biodiversity, Animal Health and Comparative Medicine, University of Glasgow, Glasgow, UK

EC, 0000-0003-2551-8589; NH, 0000-0002-1362-3499; RRK, 0000-0003-0919-6401

We introduce a game inspired by the challenges of disease management in livestock farming and the transmission of endemic disease through a trade network. Success in this game comes from balancing the cost of buying new stock with the risk that it will be carrying some disease. When players follow a simple memory-based strategy we observe a spontaneous separation into two groups corresponding to players with relatively high, or low, levels of infection. By modelling the dynamics of both the disease and the formation and breaking of trade relationships, we derive the conditions for which this separation occurs as a function of the transmission rate and the threshold level of acceptable disease for each player. When interactions in the game are restricted to players that neighbour each other in a small-world network, players tend to have similar levels of infection as their neighbours. We conclude that success in economic-epidemiological systems can originate from misfortune and geographical circumstances as well as by innate differences in personal attitudes towards risk.

## 1. Introduction

Theoretical investigation into how diseases spread has been critical to our understanding of how to control outbreaks. In the past, mathematical models and agent-based simulations have provided essential insight into the relationship between human behaviour

and pathogen biology. Examples include the effect of contact heterogeneity on the basic reproductive number of sexually transmitted diseases [1], the effect of network topology on the the critical transmissibility at which contagions thrive [2] and the conditions required for herd immunity to occur [3].

New questions emerge when we consider livestock diseases that spread over large distances when animals are traded from one farm to another [4]. In many cases surveillance and strict disease control measures are enforced to prevent epidemics from occurring, although this does not always guarantee eradication and many diseases persist in the population. Control of such *endemic* diseases therefore depends on action being taken by individual farmers [5]. This prompts many questions about risk-taking behaviour, and how much farmers are willing to pay to avoid bringing disease onto their farms [6]. The motivating question of the present work is this: What can be done to influence farmers to behave in a way that leads to the eradication of an endemic disease?

When farmers react to a disease by changing their buying behaviour, the network of potential transmission routes changes [7,8]. We therefore require a model that reflects this interplay between contagion over the links of the network, and a network continuously evolving in response to the disease. Such coevolution models have typically focused on rewiring network links away from infected nodes towards those in a healthier state [9]. The livestock disease problem is distinctly different, as individuals may be willing to accept some level of risk to avoid paying a higher price for a disease-free animal [10].

It is worth noting modelling work that concerns vaccination coverage and disease spread. When most people are vaccinated, disease prevalence is zero and the risk is perceived to be very low. For some this removes the incentive to vaccinate, reducing vaccine uptake and leading to re-emergence of the disease [11]. This dynamic has motivated a number of mathematical models, some incorporating the effects of financial cost, social influence and network structure [12–14], and commonly using a statistical physics-based modelling framework [15]. This class of models has recently expanded beyond simple vaccination to encompass a nuanced range of public health concerns [16,17].

As with vaccination, we expect farmers to be more risk averse when they are more exposed to the disease [18]. Crucially, we also expect money to play a role; for the right price some may be willing to take on a little more risk [19]. This is particularly true when we consider endemic livestock diseases. Unlike the more catastrophic epidemic threats, endemic disease is perceived by some farmers as a tolerable loss to farm productivity and animal welfare; a hard-to-avoid management problem that comes and goes in varying amounts as disease-carrying livestock are continually imported in and exported away to keep the enterprise commercially functional [20]. The novelty of this situation, when framed as a problem of the 'vaccination game' type, is the motivation behind this paper.

The challenge here is to reduce this problem to a set of rules and assumptions that capture the interplay between perceived cost–benefit motivations, the dynamics of the network and the spread of disease. One way to achieve this is to frame the problem as a game (for example [21]). In this paper we introduce a game played between a number of people that recreates the kind of dilemma associated with trading livestock in the real world. The game assumes that trading is an unavoidable component of a livestock production sector, and so of the livelihood of individual farmers. Each individual must make a purchase every turn. They cannot protect themselves through isolation or biosecurity (as in typical vaccination games) and instead must decide which other individuals they wish to be exposed to through trade.

We describe one simple memory-based strategy for playing the game and explore the dynamics of the network structure and disease outcomes when all players follow it. We observe a spontaneous divergence of disease status that splits the players into two distinct groups. In §3 we derive an expression that describes the conditions of the system that give rise to this divergence. In §3e we simulate the game on a small-world substrate network and observe that homophily between the different behaviour types emerges when spatial structure dominates.

## 2. Model

In our game there are $N$ players. Each player has some resource that they trade with other players for tokens, each unit of resource may be sick or healthy. We define the *sickness*, denoted $x_i(t)$, to be the probability that the resource traded by player $i$ is sick ($t \in \{1, 2, \ldots, T\}$ denotes the round of the game). In each round, each player privately chooses one other player. Two exchanges then occur:

> **Exchange of tokens**: every player $i$ passes one token to $j$ where $j$ is the player chosen by $i$ unless $i$ is the *only* player that chose $j$, in which case they pass none.
>
> **Exchange of sickness**: each player $i$ increases their sickness by $\beta x_j(t)$ where $j$ is the player chosen by $i$, while $j$ reduces their their sickness by the same amount. (To avoid $x_j(t+1) < 0$, if $k_j > 1/\beta$, where $k_j$ is the number of players choosing $j$, then each of them takes $x_j(t)/k_j$ from $j$.)

At the beginning of the game $x_i(0) = 1/2$ for all $i$. If the sickness level of a player exceeds a specific value then they lose the game. The objective of the game is to have the most tokens at the end of the final round.

The game is an abstract representation of the livestock trade system. The exchange of tokens incorporates a simple mechanism of supply and demand into the model; losses (in terms of tokens) are incurred when a player encounters other players wanting to buy from the same source as them. The rule that token exchange does not occur when the seller is uniquely chosen by the buyer creates an incentive to choose from players who are unlikely to be chosen by others.

Sickness may represent any of a number of diseases or physical problems that decrease the quality of the resources being traded. It is assumed here that a relatively small number of sick resources is sustainable, but when this number crosses a particular threshold the result is catastrophic to the individual. Note that the total sickness in the system does not change from one round to the next. In the case of an endemic disease that is maintained by a combination of within-group spread and trading, this 'conservation law' is a reasonable approximation of a scenario with constant prevalence. We assume no births, deaths, infections or recoveries and that sick animals simply follow a diffusion-like process through the system.

### (a) Strategy

Before introducing the strategy we take a moment to consider how a rational person might approach this game. For simplicity we assume that players do not communicate with each other in any way other than through the exchanges described in the rules of the game. All they know is what choices they made in previous rounds and the exchanges of sickness and tokens that resulted. They will therefore be able to estimate with reasonable accuracy the sickness levels of the players they traded with in previous rounds (they do not know exactly how many others chose the same as them). The remaining players will have sickness levels distributed around the population mean. Through their success or failure in retaining their tokens they will also have some indication of the popularity of the individuals they previously chose.

Now imagine that you are playing the game. Which of the $N - 1$ other players will you choose to buy from? First if retaining tokens is your priority then you will aim to choose a player that nobody else does, since this minimizes the cost of buying new animals. Disregarding any sickness concerns completely, we are left with a version of the minority game [22], a game which is won by choosing the less popular of two options, and has no rational solution (this has been adapted to allow more than two options [23]).

The additional incentive to avoid sickness provides a more stable criteria on which players can base their decisions. For example, some players may prioritize their health and aim to maintain a low sickness level. They will try to avoid players with relatively large $x_i$. Consequently the high sickness individual becomes more attractive to players who prioritize their wealth. More

generally there is a trade-off for each player between the amount of health they are willing to sacrifice and the amount of wealth they aim to accumulate.

We want to capture the essence of the health versus wealth trade-off in a strategy that can be automated. We have chosen the following strategy for its simplicity and mathematical tractability:

> At $t = 1$, the player $i$ chooses any player $j \neq i$ randomly with equal probability. For $t > 1$, supposing $i$ chose $j$ in round $t - 1$ and $k_j(t - 1)$ is the total number of players that chose $j$, $i$ will choose $j$ again if, and only if, $k_j(t-1) + x_j(t-1) < 1 + \alpha$ where $\alpha \in (0, 1)$ is a constant. Otherwise they choose randomly.

From the point of view of the player, they will repeat their choice from the previous round if that round was sufficiently successful. For this to be the case two things must be true: they choose a player that no other player chooses, and the sickness level of that player is less than a threshold $\alpha$.

This is a reasonable strategy for a player who has a very short memory. Having saved a token in the previous round it seems sensible to choose the same individual in the next one (given that the sickness level is acceptable). We note that this logic does in fact hold true in the case when every player on the game follows this same strategy. In this case the player who was selected previously will not be a target for the fraction of players who repeat their choice from the previous round, whereas a random player could be a target for both random and repeated choices.

The parameter $\alpha$ can be interpreted as the maximum amount of sickness that players are willing to accept in exchange for the increased probability of saving a token. Larger values of $\alpha$ represent a higher value being placed on the accumulation of wealth at the expense of also accumulating sickness at a faster rate. Lower values of $\alpha$ correspond to higher priority being placed on health rather than wealth.

## (b) Spatial constraint network

There are many factors that influence the decisions made by those involved in the trade of livestock. One that we wish to explore through this game is the effect of geographical constraints. We consider the players to be located on nodes of a network and restrict the choices available to each player to those with whom they share a connection. In the basic set-up the game is played on a complete network, meaning that there is an edge between every pair of individuals. In general we can use any network structure.

In §3e we limit our selection of networks to those generated by the Watts–Strogatz model [24]. Networks are constructed by placing $N$ nodes on a circle with the same distance between each pair of neighbours. Edges are created between each node and the $k$ nearest nodes in the clockwise direction. Each of these edges is then rewired to a random node with probability $p$. This creates a spatially embedded network in which nodes have a location and a mixture of short and long distance connections. By varying the parameters $k$ and $p$ we are able to generate a range of networks with varying connectivity (i.e. the mean degree $2k$) and spatial embedding (i.e. the number of connections that are determined by proximity $1 - p$). Parameter ranges are shown in table 1.

## 3. Results

We are interested in the network of trades that occur and the movement of sickness between players of the game. We focus on deriving the degree distribution of the trade network and the distribution of sickness. It is apparent that there will be a categorical difference between players with sickness level $x_i < \alpha$ and those with $x_i \geq \alpha$. We therefore consider the degree distribution and sickness distribution for the two cases separately before considering the rate at which players move from one group to the other.

We start by defining some useful concepts.

**Table 1.** Parameters and variables.

| name | notation | description | range |
|---|---|---|---|
| population | $N$ | number of players | $N > 1$ |
| tolerance | $\alpha$ | tolerance to receiving sick resources | $0 < \alpha < 1$ |
| transmission rate | $\beta$ | proportion of sickness transferred | $0 < \beta < 1$ |
| connectivity | $2k$ | mean degree of the spatial network | $2k \geq 2$ |
| randomized edges | $p$ | fraction of randomized edges in spatial network | $0 \leq p \leq 1$ |
| sickness | $x_i(t)$ | probability that $i$ provides a sick resource at time $t$ | $x_i > 0$ |
| out-degree | $k_i(t)$ | number of times $i$ chosen by others in round $t$ | $k_i \geq 0$ |
| rewire proportion | $r$ | expectation of the number of edges rewired each round | $0.56 \lesssim r \leq 1$ |
| high sickness proportion | $\lambda$ | proportion of players without capacity to retain edges $c = 0$ | $0 < \lambda < 1$ |

**Nodes and edges**: We treat each player as a node, $i$, in a dynamic network, edges in the network are directed; if $i$ chooses $j$ in one round of the game then there is an edge from $j$ to $i$.

**Out-going degree**: The number of edges going from $i$ to any other node at the end of round $t$ of the game is its out-going degree, $k_i(t)$, which we shall refer to simply as degree. Note that the in-coming degree is always 1.

**Capacity**: We say that the capacity $c_i$ of node $i$ is the number of out-going edges it can retain from one round to the next. This will be $c_i = 0$ if $x_i > \alpha$ and $c_i = 1$ otherwise. We use $n_c$ to denote the number of nodes that have capacity $c$.

**Rewiring**: We use this term to describe the change in position of an edge from one round to the next.

In this section we derive an expression for the distribution of sickness values over the population of players. There are several stages to this: we start by deriving the number of rewired edges in a typical round, we then find the degree distribution, and finally derive the sickness distribution.

## (a) Proportion of rewired edges

We use $R_t$ to denote the number of edges that are free to rewire in round $t$. A rewired edge in round $t + 1$ is created when a rewired edge at round $t$ chooses to connect to a node with capacity $c = 1$ that has retained an edge from the previous round. There are $N - R_t$ nodes of this type, and the probability that one gets chosen by at least one of the randomly rewired edges is $[1 - (1 - 1/N)^{R_t}]$. A rewired edge in round $t$ is destroyed when it chooses to rewire to a node with capacity $c = 1$ that has not retained an edge from the previous round. The number of nodes of this type is found by subtracting the nodes that have retained an edge from the previous round from the total number of nodes with capacity $c = 1$, giving $R_t - n_0$. The probability that one of these is chosen by exactly one of the randomly rewired edges is $(1/N)(1 - 1/N)^{R_t - 1}$.

We use $\langle R_t \rangle$ to denote the mean value of $R_t$ we would expect to find over a large number of realizations of the model. The change in this value over one round of the game can be expressed

$$\langle R_{t+1} \rangle - \langle R_t \rangle = (N - \langle R_t \rangle) \left[ 1 - \left( 1 - \frac{1}{N} \right)^{\langle R_t \rangle} \right] - (\langle R_t \rangle - n_0) \langle R_t \rangle \left( \frac{1}{N} \right) \left( 1 - \frac{1}{N} \right)^{\langle R_t \rangle - 1}. \quad (3.1)$$

The first term on the right-hand side is the expected increase in $R_t$ caused by newly created rewired edges. The second term represents the decrease caused by the destruction of rewired edges.

Since $\langle R_t \rangle$ cannot diverge as $t$ increases ($R_t \leq N$ for all $t$), we assume that there exists $R$ such that $\langle R_t \rangle \to R$ for large values of $t$. This simplifies the above equation to

$$(N - R)\left[1 - \left(1 - \frac{1}{N}\right)^{R}\right] = (R - n_0)\frac{R}{N}\left(1 - \frac{1}{N}\right)^{R-1}. \tag{3.2}$$

We consider the proportion of nodes in the network that have zero capacity, $\lambda = n_0/N$, and the proportion of edges that rewire each time step, $r = R/N$. By substituting these into equation (3.2), and rearranging, we can write

$$(1 - r)\left(1 + r + \frac{r^2}{2}\right) + (1 - r)\left[\left(1 - \frac{1}{N}\right)^{-rN} - 1 - r - \frac{r^2}{2}\right] = r^2 - (1 + \lambda)r + 1. \tag{3.3}$$

One can find numerically that the second term on the left-hand side is small (less than 0.022 for any $r$), and thus removing it gives a reasonable approximation to equation (3.3). We then solve to get

$$r \approx \frac{-3 + \sqrt{9 + 8(1 + \lambda)}}{2} \tag{3.4}$$

for large values of $N$. The number of rewired edges, $r$, therefore depends only on the proportion of nodes, $\lambda$, that do not have the capacity to retain an edge from one round to the next due to their high sickness level. The proportion of rewired edges is lowest when all nodes have the capacity to retain an edge, $r \approx 0.56$, yet even in this case the majority of players change their trading partner from one round to the next.

## (b) Degree distribution

We now derive the degree distribution for the network. First, we focus only on the $rN$ free edges that are rewired from the previous round. The probability $q_k$ that exactly $k$ of these edges rewires to a given node $i$ is the probability of $k$ successes out of $rN$ in a Bernoulli process where the probability of success is $1/N$. Thus the variable $k$ follows a Binomial distribution, $B(rN, 1/N)$ and since $rN \times 1/N$ converges as $N \to \infty$ it approximately follows the Poisson distribution: $q_k = e^{-r}r^k/k!$.

Here we derive the probability, $p_c(k)$, that a node that has capacity $c$ has degree $k$. For nodes with $c = 0$, simply $p_0(k) = q_k$. For those that do have the capacity to retain an edge from the previous round, $c = 1$, we have to sum over the two possibilities, that they are retaining an edge and that they are not. We therefore have

$$p_1(0) = \frac{r - \lambda}{1 - \lambda}q_0 \quad \text{and} \quad p_1(k) = \frac{1 - r}{1 - \lambda}q_{k-1} + \frac{r - \lambda}{1 - \lambda}q_k \quad \text{for } k > 0.$$

Combining the above with $p_0(k) = q_k$ we get the full degree distribution

$$p_c(k) = \begin{cases} \dfrac{e^{-r}r^k}{k!} & \text{if } c = 0 \\[2ex] \dfrac{r - \lambda}{1 - \lambda}\dfrac{e^{-r}r^k}{k!}\left[1 + \dfrac{(1 - r)k}{(r - \lambda)r}\right] & \text{if } c = 1 \end{cases} \tag{3.5}$$

Using $\langle k^n \rangle_c = \sum k^n p_c(k)$ to denote the $n$th moment, we have

$$\langle k \rangle_c = \begin{cases} r & \text{if } c = 0 \\[2ex] r + \dfrac{1 - r}{1 - \lambda} & \text{if } c = 1 \end{cases} \tag{3.6}$$

and

$$\langle k^2 \rangle_c = \begin{cases} r(1 + r) & \text{if } c = 0 \\[2ex] \dfrac{1 + (2 - \lambda)r - (1 + \lambda)r^2}{1 - \lambda} & \text{if } c = 1 \end{cases} \tag{3.7}$$

## (c) Sickness distribution

We use $x_c(t)$ to denote the expectation of the sickness level of a node with capacity $c$. Necessarily we have that $x_0(t) > \alpha$ and $x_1(t) \leq \alpha$. The changes to these values are described by the following relationship

$$x_c(t+1) - x_c(t) = -\beta x_c(t)\langle k \rangle_c + \beta r \lambda x_0(t) + \beta(1 - r\lambda)x_1(t). \tag{3.8}$$

The first term on the right-hand side describes the expected amount of sickness that the node will pass to others. The second and third terms represent the expected amount that they will receive from others given the respective probabilities of choosing a node with $c = 1$ or $c = 0$.

By design the total sickness in the system is conserved. Thus, the mean sickness will not diverge, however, we expect that nodes with capacity to retain edges to be different in this respect to those without such a capacity. To explore this, we assume that the mean sickness level of nodes will converge to one of two values corresponding to whether they have the capacity to retain an outgoing edge, or not, i.e $x_0(t) \to \mu_0$ and $x_1(t) \to \mu_1$. In this limit, the two equations above can be reduced to $\mu_0\langle k \rangle_0 = \mu_1\langle k \rangle_1$. Combining this with the expression for the mean sickness $\langle x \rangle = \lambda\mu_0 + (1 - \lambda)\mu_1 = 1/2$ we find

$$\mu_0 = \frac{1 - \lambda r}{2(r + \lambda - 2r\lambda)} \quad \text{and} \quad \mu_1 = \frac{(1 - \lambda)r}{2(r + \lambda - 2r\lambda)}. \tag{3.9}$$

The full distribution of sickness values is not easy to obtain, however, we are able to find an expression for the $n$th moment, which we then use to calculate $\sigma_c$, the standard deviation of the sickness distribution for nodes with capacity $c$, and use a Normal approximation to the distribution.

Let $\rho_c(x, t)$ be the probability density function for the sickness level $x$ at time $t$ for individuals with capacity $c$. In general, we can find $\rho_c(x, t)$ by solving

$$\rho_c(x, t) = \int \rho_c(x', t - 1)Pr(x_t = x | x_{t-1} = x')\, dx'. \tag{3.10}$$

The right-hand side integrates over all possible states at round $t$ multiplied by the probability of mapping from that state exactly on to $x$ at the next round. In our case, this transition probability is

$$Pr(x_{t+1} = x | x_t = x') = \sum_{k=0}^{\infty} p_c(k) \left[ r\lambda\delta(x - (1 - \beta k)x' - \beta\mu_0) + r\lambda\delta(x - (1 - \beta k))x' - \beta\mu_1) \right], \tag{3.11}$$

where $\delta(\cdot)$ is the Dirac delta function. The above expression is broken down as follows: $p_c(k)$ is the probability that a node has $k$ outgoing edges, $r\lambda$ is the probability that $i$ has a free link and that it rewires to one of the nodes with $c = 0$. The sickness acquired in either case is here approximated to be the mean of the group $\mu_0$ or $\mu_1$. Substituting this into equation (3.10), solving the integral, and assuming the system converges to $\rho_c(x, t) = \rho_c(x, t - 1) = \rho_c(x)$, we get

$$\rho_c(x) = \sum_{k=0}^{\infty} p_c(k)(1 - \beta k)^{-1} \left[ r\lambda\rho_c\left(\frac{x - \beta\mu_0}{1 - \beta k}\right) + (1 - r\lambda)\rho_c\left(\frac{x - \beta\mu_1}{1 - \beta k}\right) \right]. \tag{3.12}$$

Multiplying the above expression through by $x^n$ and integrating over the full range of $x$ we have

$$\int_{-\infty}^{\infty} x^n \rho_c(x)\, dx = \sum_{k=0}^{\infty} p_c(k)(1 - \beta k)^{-1} \left[ r\lambda \int_{-\infty}^{\infty} x^n \rho_c\left(\frac{x - \beta\mu_0}{1 - \beta k}\right) dx \right.$$
$$\left. + (1 - r\lambda) \int_{-\infty}^{\infty} x^n \rho_c\left(\frac{x - \beta\mu_1}{1 - \beta k}\right) dx \right]. \tag{3.13}$$

With substitutions of the form $u = (x - \beta\mu)/(1 - \beta k)$, each integral in the expression above can be rewritten

$$(1 - \beta k)^{-1} \int_{-\infty}^{\infty} x^n \rho\left(\frac{x - \beta\mu}{1 - \beta k}\right) dx = \sum_{i=0}^{n} \binom{n}{i}(1 - \beta k)^i (\beta\mu)^{n-i} \int_{-\infty}^{\infty} u^i \rho(u)\, du. \tag{3.14}$$

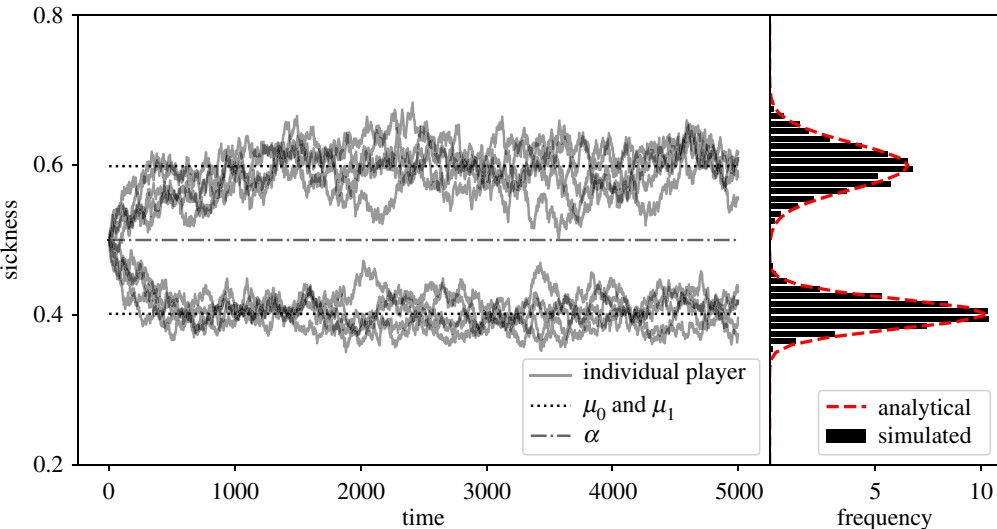

**Figure 1.** The distribution of sickness values. Simulation results for $N = 100$, $\beta = 0.05$. The left panel shows the time series for 10 randomly selected nodes and the derived mean sickness values from equation (3.9) ($\lambda$ is computed at $t = 10^3$). On the right, the frequency distribution of sickness values of all nodes at $t = 1000, 1100, \ldots, 5000$ is shown for in bins of width 0.01. The curve shows $N\rho_c(x)$ where $\rho_c(x)$ is the analytically derived Normal approximation given in equation (3.17). (Online version in colour.)

Then, using the notation $m_c(n) = \int x^n \rho_c(x)\, dx$ for the $n$th moment of $\rho_c$, the above expression becomes

$$m_c(n) = \sum_{k=0}^{\infty} p_c(k) \sum_{i=0}^{n} \binom{n}{i}(1 - \beta k)^i \left[ r\lambda(\beta\mu_0)^{n-i} + (1 - r\lambda)(\beta\mu_1)^{n-i} \right] m_c(i). \tag{3.15}$$

With $m_c(0) = 1$, we can check easily that $m_c(1) = \mu_c$. The second moment is then found to be

$$m_c(2) = \frac{[r\lambda\mu_0^2 + (1 - r\lambda)\mu_1^2]\beta + 2[r\lambda\mu_0 + (1 - r\lambda)\mu_1](1 - \beta\langle k \rangle_c)m_c(1)}{2\langle k \rangle_c - \beta\langle k^2 \rangle_c}. \tag{3.16}$$

The variance of the distribution is $\sigma_c^2 = m_c(2) - \mu_c^2$. We then use the Normal distribution

$$\rho_c(x) \approx \frac{1}{\sqrt{2\pi\sigma_c^2}} \exp\left( -\frac{(x - \mu_c)^2}{\sigma_c^2} \right) \tag{3.17}$$

as an approximation.

Figure 1 tracks the sickness level of 10 (out of 100) players over the course of the game and the mean values given analytically by equation (3.9). The curves shown in the right panel are estimates of the frequency of nodes with a given sickness level, $N \times \rho_c(x)$.

## (d) Disease status divergence

We address two questions: What proportion of nodes will go into the high sickness group? Under what values of $\alpha$ and $\beta$ does the system spontaneously divide into two distinct sickness distributions? For the first question we are able to get an approximate solution assuming the two sickness distributions are Normal and express the stability of the system as a function of $\lambda$ (the proportion of nodes that do not have capacity to retain an edge). For the second question we derive a formula from the Normal approximation that divides the parameter space into a region in which divergence is not likely to occur, and a region where it is.

The Normal approximation provides an estimate of the rate at which nodes switch from having $c = 0$ to $c = 1$ and vice versa. This switch occurs when the fluctuations in the sickness level of a

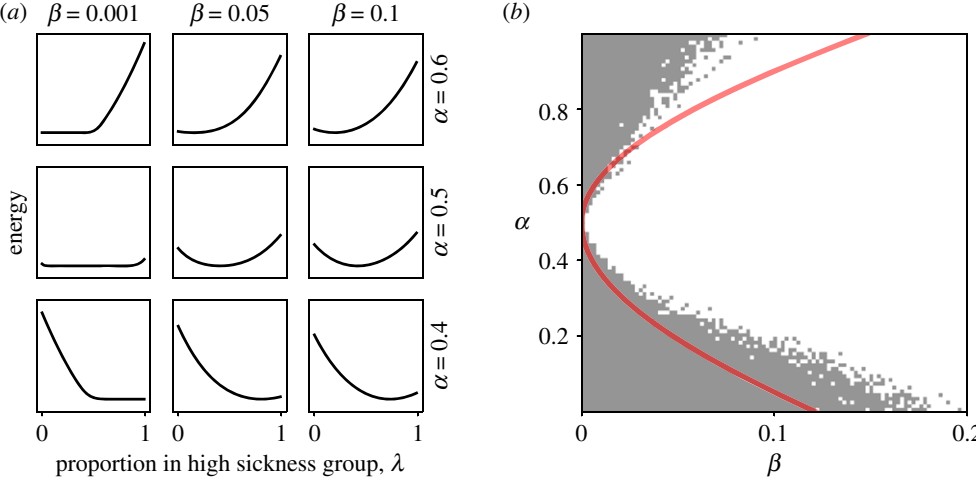

**Figure 2.** Convergence and division for different parameter combinations. (*a*) The energy landscape given by equation (3.20) for selected values of $\alpha$ and $\beta$. The system is expected to change in the direction of the slope of the curve. Local minima are therefore the stable equilibria of the system. (*b*) Regions of parameter space where divergence of disease status does or does not happen over the first 100 rounds of the game. The grey region marks where the simulation remained in its initial state, i.e. $\lambda = 1$ if $\alpha < 1/2$ or $\lambda = 0$ otherwise. The curve shows equation (3.22), the analytical approximation to this region. (Online version in colour.)

node $i$ with $c_i = 0$ ($c_i = 1$) cause $x_i$ to go below (above) $\alpha$, which is most likely to happen when the mean $\mu_0$ is only a small number of standard deviations above (below) $\alpha$. We use $\pi_{01}$ to denote our estimate of the probability that a node $i$ will change from $c_i = 0$ to $c_i = 1$. Assuming the system has converged to its steady state and the Normal approximation is valid, we have

$$\pi_{01}(\lambda) = \int_{-\infty}^{\alpha} \rho_0(x)\,\mathrm{d}x = \frac{1}{2}\left[1 + \mathrm{erf}\left(\frac{\alpha - \mu_0}{\sigma_0\sqrt{2}}\right)\right], \tag{3.18}$$

where $\mathrm{erf}(\cdot)$ is the error function. Similarly, for transitions in the opposite direction we have

$$\pi_{10}(\lambda) = \int_{\alpha}^{\infty} \rho_1(x)\,\mathrm{d}x = \frac{1}{2}\left[1 + \mathrm{erf}\left(\frac{\mu_1 - \alpha}{\sigma_1\sqrt{2}}\right)\right]. \tag{3.19}$$

In figure 2*a* we use an energy landscape to visualize the stability and convergence of the system. The potential energy of the state of the system (i.e. $\lambda\,\alpha,\,\beta$), is defined as

$$E(\lambda) = -\int_{\lambda_0}^{\lambda} l\pi_{01}(l) - (1-l)\pi_{10}(l)\,\mathrm{d}l. \tag{3.20}$$

It is a measure of the amount of external 'work' required to move the system from an arbitrary reference point $\lambda_0$ to the state $\lambda$; since the expected change from one round to the next is $\Delta(\lambda) = \lambda\pi_{01} - (1-\lambda)\pi_{10}$, the work required to increase $\lambda$ by a given arbitrarily small amount is proportional to $-\Delta(\lambda)$. The energy is then the sum of all the forces required to move from $\lambda_0$ to $\lambda$. The potential energy at different values of $\lambda$ are plotted in figure 2. Visually the line can be seen as a landscape where work is required to go uphill, and the natural course for the system is to converge to a low energy state.

At the beginning of the game all nodes will either have the capacity to retain an edge and $\lambda = 1$ (if $\alpha > 1/2$), or they will not and $\lambda = 0$ (if $\alpha \geq 1/2$). The division into two groups therefore begins as soon as one node transitions to the other which has probability

$$\pi = \frac{1}{2}\left[1 - \mathrm{erf}\left(\frac{(1 - 2\alpha)\sqrt{a - b\beta}}{\sqrt{c}\sqrt{\beta}}\right)\right] \tag{3.21}$$

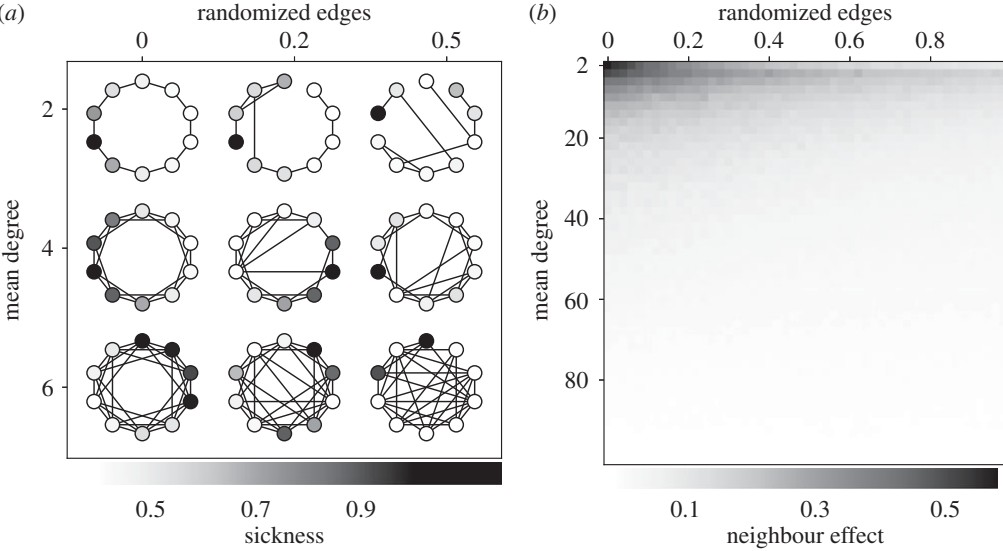

**Figure 3.** Simulation over a spatially constrained network. (*a*) Examples of the distribution of sickness after simulation of the game over small-world networks. (*b*) Each pixel shows the neighbourhood effect for a different combination of the mean degree and the proportion of long-range connections *p*. Dark regions show where the neighbourhood effect is highest implying that the disease is clustered in localized regions of the spatial network.

with the constants $a = 1$, $b = 1$ and $c = 1$ if $\alpha < 1/2$ and $a = 4$, $b = 5\sqrt{17} - 17$, $c = 10\sqrt{17} - 38$ if $\alpha \geq 1/2$.

The probability that division does not occur in say $T$ rounds of the game is approximated by $p = (1 - \pi)^{NT}$; this assumes each round independently draws from the appropriate Normal distribution; however, this will be an overestimate since the variance starts at 0 in the first round and converges to its steady-state value $\sigma_c^2$. For a given value of $p$, it follows from the above equation that division will occur in the first $T$ rounds of the game with probability $p$ if

$$\beta > \frac{a}{b + c(z/1 - 2\alpha)^2}, \tag{3.22}$$

where $z = \mathrm{erf}^{-1}(2p^{1/NT} - 1)$.

Shown in figure 2 are the results from simulations of 100 rounds of the game. We also show equation (3.21) for $p = 1/2$, to indicate the region where divergence of disease status is more likely to happen than not. The Normal distribution overestimates the probability density in the left tail and underestimates it in the right tail. Thus, $\pi_{01}(0)$ is larger than it would be if the approximation was more accurate, and so in the case where $\alpha < 1/2$, divergence is predicted to occur for parameter combinations that do not diverge in the simulation. Similarly, when $\alpha > 1/2$, $\pi_{10}(1)$ is larger than it should be and divergence occurs in places where the analytical result suggests it will not.

## (e) Spatial constraint network

We ask how spatial constraints on the choices of the players affect the distribution of sickness values. Figure 3*a* shows some examples of the networks generated using the Watts–Strogatz model. The disease simulation was performed on the network for 100 rounds with $\alpha = 0.5$ and $\beta = 0.05$ and the resulting sickness values of each node are also shown. The examples shown demonstrate that sickness is most concentrated in localized regions of the spatial network when edges are more constrained; nodes with high levels of sickness tend to be connected to others with high levels of sickness.

Figure 3*b* shows this *neighbourhood effect* between sickness more formally. Here each pixel represents a different combination of *p* and *k*. A network was generated for each combination, the simulation was then performed on the network and the slope of the least-squares regression between the sickness of a node and the mean sickness of its neighbours calculated. This was repeated 100 times and the mean taken.

As the mean degree increases (increasing the number of options available to each player) the correlation between the sickness of neighbours decreases until a nodes location does not have any effect on its sickness value. We also see that when choice is highly constrained (low *p* and low *k*), the addition of a small number of long-range (rewired) edges can be effective in mitigating this neighbourhood effect.

## 4. Discussion

We have described a game inspired by the challenges of livestock management where the financial cost of disease is a major incentive to take precautions when buying. Through the dynamics of the game, players encounter situations where they must decide between prioritizing wealth over health, bringing them closer to catastrophic levels of sickness, or prioritizing their health over wealth and gaining less financially.

When all players follow a simple strategy, only changing their choice from the previous round if it was detrimental to their health or their wealth, the population is likely to separate into two distinct groups: one with a relatively high level of sickness compared with the other. Whether or not the separation occurs depends on two parameters: the amount of disease that individual players consider to be excessive and the amount of sickness transmitted in one transaction.

Players in one group are identical to those in the other regarding how they make decisions; the divergence happens because the disease status of a player affects how they are *perceived* by others. After a transaction, the buyer learns the sickness level of the seller and decides whether to return to them in the next round. This results in a kind of virtuous (or vicious) cycle that drives the divergence of the groups. Players with good health status are rewarded by being chosen more frequently, they transmit more of their sickness up to the less healthy group, while those in the high sickness group are chosen less frequently causing the outward flow of sickness to be slower in comparison.

One might suspect that the configuration of edges in the first round determines the fate of each player but this is not always the case; under some circumstances the sickness distribution remains unimodal for some time before random fluctuations drive it towards a more stable bimodal state. The idea that structure can emerge from a homogeneous starting configuration without any external forces shaping it is known as symmetry breaking, and is associated with spontaneous pattern formation in biological systems [25].

The model provides a basis for exploring the role of space and network structure on contagion dynamics. In addition to the possibility that random chance fluctuations, rather than players' strategy, can determine the fate of the population, we have seen also that the location of the player within a spatially constrained network can also be a factor. This is due to a neighbourhood effect and the fact that in this model the total amount of disease is conserved. As the sickness begins to aggregate in a particular region of the network, those within that region become exposed to a greater number of high sickness options and fewer healthy players to choose from. While long-range connections typically increase the risk of an epidemic (e.g. [7]), in this model they mitigate the neighbourhood effect, which helps avoid extreme levels being reached by any one player.

Returning to the motivating question, what can be done to influence farmers to behave in a way that leads to the eradication of an endemic disease? In our model we found that a small change in $\alpha$ can potentially make a dramatic difference in how disease is distributed across the population. Recall that this parameter represents the maximum amount of sickness players will tolerate before deciding to look elsewhere for a better trading relationship. Our results strengthen the argument that small differences in the general attitude of the farming community towards endemic disease can make a large difference to its prevalence.

An obvious limitation of the strategy we have chosen to investigate is that very little information is used by the players. More complex strategies could be explored that use the history of choices and outcomes, such as those that have been applied to the minority game [22,26]. These strategies are allowed to evolve over time with mutations, reproduction and extinction, analogous to biological evolution [27]. Decisions could also be informed by information about other players. If the game was played over a table, for example, players will see the trades being made by others and be able to infer second order information from the activity they observe. In the context of livestock trade it is not known exactly how information spreads, however, we suspect that the networks over which information transmits will strongly influence the pattern of trading and the progression of endemic diseases [28,29].

This work provokes ideas that could potentially be tested either experimentally or through observation, directly, or in data such as the cattle movement databases that now exist across many countries. Analysis of the decisions made by human players could be compared with more traditional measures of risk taking [30]. Discrete choice experiments, for example, offer a way to assess the willingness of individuals to pay for livestock of differing levels of health [31,32].

The challenge, ultimately, is to find ways to control and eradicate endemic disease by influencing the behaviour of individuals. Our results show that this may be possible. Nudges towards more congenial behaviour can be magnified by the feedback loop between behaviour and disease spread. Exploiting this effect can push the system towards a stable equilibrium that yields a better payoff for the population as a whole.

Data accessibility. No data were used. All scripts used to generate figures are available at https://github.com/EwanColman/network-rewiring-game.

Authors' contributions. E.C., N.H. and R.R.K. conceived the study and conducted analysis. E.C. drafted the manuscript. R.R.K. and N.H. revised and edited the manuscript. All the authors have approved the publication of this manuscript and agreed to be accountable for the accuracy or integrity of its content.

Competing interests. We declare we have no competing interests.

Funding. This work was funded by Wellcome Trust grant no. 209818/Z/17/Z.

Acknowledgements. We would like to thank the members of the Farm-level Interdisciplinary approaches to Endemic Livestock Disease (FIELD) project for their support and feedback. We are also grateful for the work of the two anonymous reviewers and the associate editor.

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
