## [Reviewer comments · Proceedings. Mathematical, Physical, and Engineering Sciences]

Review History

RSPA-2019-0837.R0 (Original submission)

Review form: Referee 1

Is the manuscript an original and important contribution to its field?

Acceptable

Is the paper of sufficient general interest?

Acceptable

Is the overall quality of the paper suitable?

Acceptable

Can the paper be shortened without overall detriment to the main message?

Yes

Do you think some of the material would be more appropriate as an electronic appendix?

No

Do you have any ethical concerns with this paper?

No

Recommendation?

Major revision is needed (please make suggestions in comments)

Comments to the Author(s)

The authors develop a game to simulate the transmission of an infectious disease on a network where traders choose between health and wealth.

The manuscript is interesting, well developed and well written. However, many assumptions in developing the game and in the analysis need to be better clarified before publication.

Specific comments

- 1) There needs to be a better clarification and description of the interpretation of the game. Understandably any games described need to be a simplification of real world settings. However, the dynamics and exchanges between traders in the game are not clearly related to their motivations in the real world. In the game, the players are essentially trading tokens for sickness. What is their primary motivation for trading? How does that relate to real world markets?
- 2) If only one player chooses another and there is no movement of tokens, is sickness still exchanged?
- 3) The last sentence in the first paragraph of section 2 (on page 2) states that after a player has made a decision, choices are revealed to everyone. This contradicts the paragraph of the Strategy subsection (on page 3).
- 4) How is sickness initially distributed across the nodes?
- 5) The authors should provide a table with a description of all parameters.
- 6) On page 5, how reasonable is the assumption that there is an equilibrium solution that $\$R_t\$$ approaches for large values of $\$t\$$? This needs to be better justified.
- 7) Isn't the Taylor approximation given above equation (3.3) for $(1-1/N)^R$ and not for $(-R)$ in the exponent?
- 8) The authors should provide a better derivation and justification for (3.3) - perhaps in an appendix? How large does N need to be for this to hold? The expression provides a fairly narrow range for $\$r\$$? What does this imply for the game?
- 9) In the second paragraph of (c), the authors state that they expect the mean sickness levels of nodes to converge to one of two values. Again - how reasonable is this assumption? In section (e), the author address the question of simulations diverging to 2 separate nodes - but such justifications need to be provided before such assumptions, including clarifying the ranges of parameter and ranges where they hold.

The analysis in (c) and Figure 1 are built on these equilibrium assumptions without justification. Could the authors also simulate the game without these assumptions to see if the results still hold? Could they provide a figure like Figure 1 starting from the description of the game in Section 2 with no additional equilibrium assumptions?

Review form: Referee 2

Is the manuscript an original and important contribution to its field?

Good

Is the paper of sufficient general interest?

Good

Is the overall quality of the paper suitable?

Good

Can the paper be shortened without overall detriment to the main message?

Yes

Do you think some of the material would be more appropriate as an electronic appendix?

No

Do you have any ethical concerns with this paper?

No

Recommendation?

Accept with minor revision (please list in comments)

Comments to the Author(s)

This work tries to pose a quite new type of ‘vaccination game’. When one calls a vaccination game, it implies models based on mathematical epidemiology; such as SIS, SIR and so forth, to depict disease spreading process, combined with the framework of evolutionary game theory to reproduce an individual decision whether committing a vaccination or not depending on the balance between vaccination cost and potential risk to be infected unless vaccinated (aiming ‘free-ride’ to the so-called herd immunity). As well-known, there have been quite a few vaccination game (VG) models in the sense of conventional VG abovementioned. What the authors establishing here has an entirely different scope. In this sense, it is said quite new. I would say it is an admirable trial, although I see it quite specific and less universal vis-à-vis usual VG models. The game framework is depicted twofold relations of agents (players) that are connected underling network that may be time-evolving (taking place refreshments of a random link like dynamic small world network) or time-frozen. The twofold relations are what they are called; ‘exchange of money’ (more fairly to be said ‘resource’) and ‘exchange of sickness’. A game proceeds with each of agent, obeying to his/ her strategy exchanging resource. On the process of such exchange, disease spreads. Quite interestingly (or strangely) to me, when an agent successfully passes a unit of sickness to a contact person, the contact person may be infected. That’s fine, since it implies a transmittance process between I and S states. Yet, the focal agent who gives such unwilling gift to the other person is able to successfully lose a unit of sickness. I’m not so sure what this idea implying in relation with what’s happening in real world. Base on such assumptions, the authors established the mathematical model, which seems scientifically robust. In fact, they drew analytical solutions, and, interestingly, they found that there are bi-stable-like equilibrium; lower sickness (μ_0) and higher one (μ_1), given in Eq. (8). Fig. 1 visually confirms how sickness time-evolves, which seems interesting. Fig. 2 gives little bit more profound discussion about how higher and lower sickness states bifurcate in relation with two model parameters; Alpha (central level of those two) and Beta (disease transmittance rate).

In the very last part, the authors discussed what it would be if underling network is spatially constrained of which result is summarized in Fig. 3.

In sum, I see this work successfully presenting highly reliable results in terms of science. Thus, basically I embrace a positive feeling. Again, I agree that the model presented here is quite new. Yet, I think that it is quite different from the conventional VG models. To make this MS more impressive to the audience who is interested in VG models, I think the authors should more carefully and deliberately highlight how their model is new and different from the usual VG models by citing some those representative (and quite recent ones), for instance;

Wang, Bauch, Bhattacharyya, d'Onofrio, Manfredi, Perc, et al., Statistical physics of vaccination, *Physics Report* 664, 1-133, 2016.

Arefin et al.; Interplay between cost and effectiveness in influenza vaccination uptake: vaccination game approach, *Proceedings of the Royal Society A* 475, 20190608, 2019.

Kabir et al.; Modelling and analysing the coexistence of dual dilemmas in the proactive vaccination game and retroactive treatment game in epidemic viral dynamics, *Proceedings of the Royal Society A* 475, 20190484, 2019. <https://doi.org/10.1098/rspa.2019.0484>

Kabir et al.; Behavioral incentives in a vaccination-dilemma setting with optional treatment, *Physical Review E* 100, 062402, 2019.

Evolutionary Games with Sociophysics: Analysis of Traffic Flow and Epidemics, Springer, 2019.1; ISBN: 978-981-13-2769-8

Fundamentals of Evolutionary Game Theory and its Applications, Springer, 2015.10; ISBN: 978-4-431-54961-1

Decision letter (RSPA-2019-0837.R0)

04-Mar-2020

Dear Dr Colman

The Editor of *Proceedings A* has now received comments from referees on the above paper and would like you to revise it in accordance with their suggestions which can be found below (not including confidential reports to the Editor).

Please submit a copy of your revised paper within four weeks - if we do not hear from you within this time then it will be assumed that the paper has been withdrawn. In exceptional circumstances, extensions may be possible if agreed with the Editorial Office in advance.

Please note that it is the editorial policy of *Proceedings A* to offer authors one round of revision in which to address changes requested by referees. If the revisions are not considered satisfactory by the Editor, then the paper will be rejected, and not considered further for publication by the journal. In the event that the author chooses not to address a referee's comments, and no scientific justification is included in their cover letter for this omission, it is at the discretion of the Editor whether to continue considering the manuscript.

- Acknowledgements
- Funding statement

To revise your manuscript, log into <http://mc.manuscriptcentral.com/prsa> and enter your Author Centre, where you will find your manuscript title listed under "Manuscripts with Decisions." Under "Actions," click on "Create a Revision." Your manuscript number has been appended to denote a revision.

You will be unable to make your revisions on the originally submitted version of the manuscript. Instead, revise your manuscript and upload a new version through your Author Centre.

When submitting your revised manuscript, you will be able to respond to the comments made by the referee(s) and upload a file "Response to Referees" in "Section 6 - File Upload". Please use this to document how you have responded to the comments, and the adjustments you have made. In order to expedite the processing of the revised manuscript, please be as specific as possible in your response to the referee(s).

IMPORTANT: Your original files are available to you when you upload your revised manuscript. Please delete any unnecessary previous files before uploading your revised version.

When revising your paper please ensure that it remains under 28 pages long. In addition, any pages over 20 will be subject to a charge (£150 + VAT (where applicable) per page). Your paper has been ESTIMATED to be 12 pages.

Once again, thank you for submitting your manuscript to Proc. R. Soc. A and I look forward to receiving your revision. If you have any questions at all, please do not hesitate to get in touch.

Yours sincerely
Raminder Shergill
proceedingsa@royalsociety.org

Reviewer(s)' Comments to Author:

Referee: 1

Comments to the Author(s)

The authors develop a game to simulate the transmission of an infections disease on a network where traders choose between health and wealth.

The manuscript is interesting, well developed and well written. However, many assumptions in developing the game and in the analysis need to be better clarified before publication.

Specific comments

- 1) There needs to be a better clarification and description of the interpretation of the game. Understandably any games described need to be a simplification of real world settings. However, the dynamics and exchanges between traders in the game are not clearly related to their motivations in the real world. In the game, the players are essentially trading tokens for sickness. What is their primary motivation for trading? How does that relate to real world markets?
- 2) If only one player chooses another and there is no movement of tokens, is sickness still exchanged?
- 3) The last sentence in the first paragraph of section 2 (on page 2) states that after a player has made a decision, choices are revealed to everyone. This contradicts the paragraph of the Strategy subsection (on page 3).
- 4) How is sickness initially distributed across the nodes?
- 5) The authors should provide a table with a description of all parameters.
- 6) On page 5, how reasonable is the assumption that there is an equilibrium solution that $\$R_t\$$ approaches for large values of $\$t\$$? This needs to be better justified.
- 7) Isn't the Taylor approximation given above equation (3.3) for $(1-1/N)^R$ and not for $(-R)$ in the exponent?

8) The authors should provide a better derivation and justification for (3.3) - perhaps in an appendix? How large does N need to be for this to hold? The expression provides a fairly narrow range for β ? What does this imply for the game?

9) In the second paragraph of (c), the authors state that they expect the mean sickness levels of nodes to converge to one of two values. Again - how reasonable is this assumption? In section (e), the author address the question of simulations diverging to 2 separate nodes - but such justifications need to be provided before such assumptions, including clarifying the ranges of parameter and ranges where they hold.

The analysis in (c) and Figure 1 are built on these equilibrium assumptions without justification. Could the authors also simulate the game without these assumptions to see if the results still hold? Could they provide a figure like Figure 1 starting from the description of the game in Section 2 with no additional equilibrium assumptions?

Referee: 2

Comments to the Author(s)

This work tries to pose a quite new type of 'vaccination game'. When one calls a vaccination game, it implies models based on mathematical epidemiology; such as SIS, SIR and so forth, to depict disease spreading process, combined with the framework of evolutionary game theory to reproduce an individual decision whether committing a vaccination or not depending on the balance between vaccination cost and potential risk to be infected unless vaccinated (aiming 'free-ride' to the so-called herd immunity). As well-known, there have been quite a few vaccination game (VG) models in the sense of conventional VG abovementioned. What the authors establishing here has a entirely different scope. In this sense, it is said quite new. I would say it is an admirable trial, although I see it quite specific and less universal vis-à-vis usual VG models. The game framework is depicted twofold relations of agents (players) that are connected underling network that may be time-evolving (taking place refreshments of a random link like dynamic small world network) or time-frozen. The twofold relations are what they are called; 'exchange of money' (more fairly to be said 'resource') and 'exchange of sickness'. A game proceeds with each of agent, obeying to his/ her strategy exchanging resource. On the process of such exchange, disease spreads. Quite interestingly (or strangely) to me, when an agent successfully passes a unit of sickness to a contact person, the contact person may be infected. That's fine, since it implies a transmittance process between I and S states. Yet, the focal agent who gives such unwilling gift to the other person is able to successfully lose a unit of sickness. I'm not so sure what this idea implying in relation with what's happening in real world. Base on such assumptions, the authors established the mathematical model, which seems scientifically robust. In fact, they drew analytical solutions, and, interestingly, they found that there are bi-stable-like equilibrium; lower sickness (μ_0) and higher one (μ_1), given in Eq. (8). Fig. 1 visually confirms how sickness time-evolves, which seems interesting. Fig. 2 gives little bit more profound discussion about how higher and lower sickness states bifurcate in relation with two model parameters; α (central level of those two) and β (disease transmittance rate).

In the very last part, the authors discussed what it would be if underling network is spatially constrained of which result is summarized in Fig. 3.

In sum, I see this work successfully presenting highly reliable results in terms of science. Thus, basically I embrace a positive feeling. Again, I agree that the model presented here is quite new. Yet, I think that it is quite different from the conventional VG models. To make this MS more impressive to the audience who is interested in VG models, I think the authors should more carefully and deliberately highlight how their model is new and different from the usual VG models by citing some those representative (and quite recent ones), for instance;

Wang, Bauch, Bhattacharyya, d'Onofrio, Manfredi, Perc, et al., Statistical physics of vaccination, Physics Report 664, 1-133, 2016.

Arefin et al.; Interplay between cost and effectiveness in influenza vaccination uptake: vaccination game approach, Proceedings of the Royal Society A 475, 20190608, 2019.

Kabir et al.; Modelling and analysing the coexistence of dual dilemmas in the proactive vaccination game and retroactive treatment game in epidemic viral dynamics, Proceedings of the Royal Society A 475, 20190484, 2019. <https://doi.org/10.1098/rspa.2019.0484>

Kabir et al.; Behavioral incentives in a vaccination-dilemma setting with optional treatment, Physical Review E 100, 062402, 2019.

Evolutionary Games with Sociophysics: Analysis of Traffic Flow and Epidemics, Springer, 2019.1; ISBN: 978-981-13-2769-8

Fundamentals of Evolutionary Game Theory and its Applications, Springer, 2015.10; ISBN: 978-4-431-54961-1

Board Member:

Comments to Author(s):

Please address reviewers' points. It would be beneficial to expand on motivation and highlight again at the beginning of Model section that you are dealing with endemic diseases as opposed to epidemic (as mentioning epidemic models in the intro seem to nudge toward a wrong perception of your setting) .Further clarification of the initialisation of the game and different parameters' ranges is needed.

Author's Response to Decision Letter for (RSPA-2019-0837.R0)

See Appendix A.

RSPA-2019-0837.R1 (Revision)

Review form: Referee 1

Is the manuscript an original and important contribution to its field?

Good

Is the paper of sufficient general interest?

Good

Is the overall quality of the paper suitable?

Acceptable

Can the paper be shortened without overall detriment to the main message?

Yes

Do you think some of the material would be more appropriate as an electronic appendix?

No

Do you have any ethical concerns with this paper?

No

Recommendation?

Major revision is needed (please make suggestions in comments)

Comments to the Author(s)

The authors have addressed many of comments from the previous version. However, some of their responses do not address the comments and appear to side step around the issue (or are mathematically incorrect).

1) The authors state:

Since R_t cannot diverge as t increases ($R_t < N$ for all t), we assume that there exists R such that $R_t < R$ for large values of t .

There is still no justification for this assumption. Even if R_t cannot diverge, it may asymptotically approach periodic orbits or even exhibit chaotic dynamics. Why do they assume that there is a globally asymptotically stable equilibrium point?

2) In their response, the authors state:

The range of N is now given in Table 1. There is nothing about equation (3.3) that requires N to be very large.

I frankly don't understand the authors' response here. Equation 3.3 is derived using a Taylor approximation for r close to 0 (or N approaching infinity), as is stated in the manuscript. I repeat my previous comment.

The authors should provide a better derivation and justification for (3.3) perhaps in an appendix? How large does N need to be for this to hold?

3) In their response, the authors state, "

Yes. The absence of token exchange is a mechanism to incentivise the choice of low demand suppliers in a way that mimics real market dynamics. This is now clearly stated in terms of resources:

Each player has some resource that they trade with other players for tokens, each unit of resource may be sick or healthy.

I suggest they also add a statement to the manuscript describing why they model no exchange of token as written above..

Review form: Referee 2**Is the manuscript an original and important contribution to its field?**

Good

Is the paper of sufficient general interest?

Good

Is the overall quality of the paper suitable?

Good

Can the paper be shortened without overall detriment to the main message?

Yes

Do you think some of the material would be more appropriate as an electronic appendix?

No

Do you have any ethical concerns with this paper?

No

Recommendation?

Accept as is

Comments to the Author(s)

The revised version seems clear and appropriate for publication.

Decision letter (RSPA-2019-0837.R1)

12-Jun-2020

Dear Dr Colman

The Editor of Proceedings A has now received comments from referees on the above paper and would like you to revise it in accordance with their suggestions which can be found below (not including confidential reports to the Editor).

Please submit a copy of your revised paper within four weeks - if we do not hear from you within this time then it will be assumed that the paper has been withdrawn. In exceptional circumstances, extensions may be possible if agreed with the Editorial Office in advance.

Please note that it is the editorial policy of Proceedings A to offer authors one round of revision in which to address changes requested by referees. If the revisions are not considered satisfactory by the Editor, then the paper will be rejected, and not considered further for publication by the journal. In the event that the author chooses not to address a referee's comments, and no scientific justification is included in their cover letter for this omission, it is at the discretion of the Editor whether to continue considering the manuscript.

- Acknowledgements
- Funding statement

To revise your manuscript, log into <http://mc.manuscriptcentral.com/prsa> and enter your Author Centre, where you will find your manuscript title listed under "Manuscripts with Decisions." Under "Actions," click on "Create a Revision." Your manuscript number has been appended to denote a revision.

You will be unable to make your revisions on the originally submitted version of the manuscript. Instead, revise your manuscript and upload a new version through your Author Centre.

When submitting your revised manuscript, you will be able to respond to the comments made by the referee(s) and upload a file "Response to Referees" in "Section 6 - File Upload". Please use this to document how you have responded to the comments, and the adjustments you have made. In order to expedite the processing of the revised manuscript, please be as specific as possible in your response to the referee(s).

IMPORTANT: Your original files are available to you when you upload your revised manuscript. Please delete any unnecessary previous files before uploading your revised version.

When revising your paper please ensure that it remains under 28 pages long. In addition, any pages over 20 will be subject to a charge (£150 + VAT (where applicable) per page). Your paper has been ESTIMATED to be 13 pages.

Once again, thank you for submitting your manuscript to Proc. R. Soc. A and I look forward to receiving your revision. If you have any questions at all, please do not hesitate to get in touch.

Yours sincerely
Raminder Shergill
proceedingsa@royalsociety.org

Reviewer(s)' Comments to Author:

Referee: 2

Comments to the Author(s)
The revised version seems clear and appropriate for publication.

Referee: 1

Comments to the Author(s)
The authors have addressed many of comments from the previous version. However, some of their responses do not address the comments and appear to side step around the issue (or are mathematically incorrect).

1) The authors state:

Since R_t cannot diverge as t increases ($R_t \leq N$ for all t), we assume that there exists R such that $R_t \leq R$ for large values of t .

There is still no justification for this assumption. Even if R_t cannot diverge, it may asymptotically approach periodic orbits or even exhibit chaotic dynamics. Why do they assume that there is a globally asymptotically stable equilibrium point?

2) In their response, the authors state:

The range of N is now given in Table 1. There is nothing about equation (3.3) that requires N to be very large.

I frankly don't understand the authors' response here. Equation 3.3 is derived using a Taylor approximation for r close to 0 (or N approaching infinity), as is stated in the manuscript. I repeat my previous comment.

The authors should provide a better derivation and justification for (3.3) perhaps in an appendix? How large does N need to be for this to hold?

3) In their response, the authors state, "

Yes. The absence of token exchange is a mechanism to incentivise the choice of low demand suppliers in a way that mimics real market dynamics. This is now clearly stated in terms of resources:

Each player has some resource that they trade with other players for tokens, each unit of resource may be sick or healthy.

I suggest they also add a statement to the manuscript describing why they model no exchange of token as written above..

Board Member

Comments to Author(s):

We thank the authors for submitting a revision. As can be seen from the reviews, there are still some issues remaining for authors to address. Given the importance of the topic during the time of the pandemic, we decided to give authors another opportunity to respond to reviews. Please check the reviews for detailed comments.

Author's Response to Decision Letter for (RSPA-2019-0837.R1)

See Appendix B.

RSPA-2019-0837.R2 (Revision)

Review form: Referee 2

Is the manuscript an original and important contribution to its field?

Acceptable

Is the paper of sufficient general interest?

Acceptable

Is the overall quality of the paper suitable?

Acceptable

Can the paper be shortened without overall detriment to the main message?

Yes

Do you have any ethical concerns with this paper?

No

Recommendation?

Accept as is

Comments to the Author(s)

The revision seems enough to be accepted.

Decision letter (RSPA-2019-0837.R2)

09-Sep-2020

Dear Dr Colman

I am pleased to inform you that your manuscript entitled "Spontaneous divergence of disease status in an economic epidemiological game" has been accepted in its final form for publication in Proceedings A.

Our Production Office will be in contact with you in due course. You can expect to receive a proof of your article soon. Please contact the office to let us know if you are likely to be away from e-mail in the near future. If you do not notify us and comments are not received within 5 days of sending the proof, we may publish the paper as it stands.

Open access

You are invited to opt for open access, our author pays publishing model. Payment of open access fees will enable your article to be made freely available via the Royal Society website as soon as it is ready for publication. For more information about open access please visit <https://royalsociety.org/journals/authors/which-journal/open-access/>. The open access fee for this journal is £1700/\$2380/€2040 per article. VAT will be charged where applicable.

Note that if you have opted for open access then payment will be required before the article is published – payment instructions will follow shortly.

If you wish to opt for open access then please inform the editorial office (proceedingsa@royalsociety.org) as soon as possible.

Your article has been estimated as being 14 pages long. Our Production Office will inform you of the exact length at the proof stage.

Proceedings A levies charges for articles which exceed 20 printed pages. (based upon approximately 540 words or 2 figures per page). Articles exceeding this limit will incur page charges of £150 per page or part page, plus VAT (where applicable).

Under the terms of our licence to publish you may post the author generated postprint (ie. your accepted version not the final typeset version) of your manuscript at any time and this can be made freely available. Postprints can be deposited on a personal or institutional website, or a recognised server/repository. Please note however, that the reporting of postprints is subject to a media embargo, and that the status the manuscript should be made clear. Upon publication of the definitive version on the publisher's site, full details and a link should be added.

You can cite the article in advance of publication using its DOI. The DOI will take the form: 10.1098/rspa.XXXX.YYYY, where XXXX and YYYY are the last 8 digits of your manuscript number (eg. if your manuscript number is RSPA-2017-1234 the DOI would be 10.1098/rspa.2017.1234).

For tips on promoting your accepted paper see our blog post: <https://royalsociety.org/blog/2020/07/promoting-your-latest-paper-and-tracking-your-results/>

On behalf of the Editor of Proceedings A, we look forward to your continued contributions to the Journal.

Sincerely,
Raminder Shergill
proceedingsa@royalsociety.org

Reviewer(s)' Comments to Author:

Referee: 2

Comments to the Author(s)
The revision seems enough to be accepted.

Dear Editor of Proceedings A,

We are grateful for the efforts of both reviewers. We have made a thorough revision following their comments and believe the manuscript meets the requirements of the editor. Please see our detailed replies to the reviewers' comments below.

Kindest regards,

Ewan Colman,
Nick Hanley,
Rowland Kao

Note: The reviewers' comments are indented.

Reviewer #1:

The authors develop a game to simulate the transmission of an infectious disease on a network where traders choose between health and wealth.

The manuscript is interesting, well developed and well written. However, many assumptions in developing the game and in the analysis need to be better clarified before publication.

We are grateful to the reviewer for their encouraging words and the care that they have put into reviewing our manuscript.

1) There needs to be a better clarification and description of the interpretation of the game. Understandably any games described need to be a simplification of real world settings. However, the dynamics and exchanges between traders in the game are not clearly related to their motivations in the real world. In the game, the players are essentially trading tokens for sickness. What is their primary motivation for trading? How does that relate to real world markets?

The following passages have been added to the introduction:

Unlike the more catastrophic epidemic threats, endemic disease is perceived by some farmers as a tolerable loss to farm productivity and animal welfare; a hard-to-avoid management problem that comes and goes in varying amounts as disease carrying livestock are continually imported in and exported away to keep the enterprise commercially functional [20]. The novelty of this situation, when framed as a problem of the "vaccination game" type, is the motivation behind this paper.
And,

The game assumes that trading is an unavoidable component of a livestock production sector, and so of the livelihood of individual farmers. Each individual must make a purchase every turn. They cannot protect themselves through isolation or biosecurity (as in typical vaccination games) and instead must decide which other individuals they wish to be exposed to through trade.

References are also included to the relevant literature.

2) If only one player chooses another and there is no movement of tokens, is sickness still exchanged?

Yes. The absence of token exchange is a mechanism to incentivise the choice of low-demand suppliers in a way that mimics real market dynamics. This is now clearly stated in terms of resources:

Each player has some resource that they trade with other players for tokens, each unit of resource may be sick or healthy.

while the second paragraph of the model section (unchanged) justifies the reason for the exchange mechanism

3) The last sentence in the first paragraph of section 2 (on page 2) states that after a player has made a decision, choices are revealed to everyone. This contradicts the paragraph of the Strategy subsection (on page 3).

That is true. On reflection we don't think see any reason why we need to say that the choices are revealed - they can remain private. We have removed that sentence.

4) How is sickness initially distributed across the nodes?

It was our error to leave out that important detail. In the model section we have now added the following in the first part of the model section.

At the beginning of the game $x_{i}(t)=1/2$ for all i .

5) The authors should provide a table with a description of all parameters.

We have added Table 1 containing parameters and important variables.

6) On page 5, how reasonable is the assumption that there is an equilibrium solution that R_t approaches for large values of t ? This needs to be better justified.

We have added the following details to the part where this is mentioned:

Since R_t cannot diverge as t increases ($R_t \leq N$ for all t), we assume that there exists R such that $R_t \rightarrow R$ for large values of t .

7) Isn't the Taylor approximation given above equation (3.3) for $(1-1/N)^R$ and not for $(-R)$ in the exponent?

Yes. It has now been corrected.

8) The authors should provide a better derivation and justification for (3.3) - perhaps in an appendix? How large does N need to be for this to hold?

Have moved most of the explanation to before the equation and reworded it to be clearer. It now reads,

We use R_t to denote the expected number of edges that are free to rewire in round t . A rewired edge in round $t+1$ is created when a rewired edge at round t chooses to connect to a

node with capacity $c=1$ that has retained an edge from the previous round. There are $N-R_{t-1}$ nodes of this type, and the probability that one gets chosen by at least one of the randomly rewired edges is $[1-(1-1/N)^{R_{t-1}}]$. A rewired edge in round t is destroyed when it chooses to rewire to a node with capacity $c=1$ that has not retained an edge from the previous round. The number of nodes of this type is found by subtracting the nodes that have retained an edge from the previous round from the total number of nodes with capacity $c=1$, giving $R_{t-1}-n_{0,t}$. The probability that one of these is chosen by exactly one of the randomly rewired edges is $(1/N)(1-1/N)^{R_{t-1}-1}$.

The range of N is now given in Table 1. There is nothing about equation (3.3) that requires N to be very large.

| The expression provides a fairly narrow range for r ? What does this imply for the game?

The following has been added:

The proportion of rewired edges is lowest when all nodes have the capacity to retain an edge, $r \approx 0.56$, yet even in this case the majority of players change their trading partner from one round to the next.

| 9) In the second paragraph of (c), the authors state that they expect the mean sickness levels of nodes to converge to one of two values. Again - how reasonable is this assumption?

Before this part we have added:

By design the total sickness in the system is conserved. Thus, the mean sickness will not diverge, however we expect that nodes with capacity to retain edges to be different in this respect to those without such a capacity.

| In section (e), the author address the question of simulations diverging to 2 separate nodes - but such justifications need to be provided before such assumptions, including clarifying the ranges of parameter and ranges where they hold.

It is not clear to us what the reviewer is referring to in this comment.

| The analysis in (c) and Figure 1 are built on these equilibrium assumptions without justification. Could the authors also simulate the game without these assumptions to see if the results still hold? Could they provide a figure like Figure 1 starting from the description of the game in Section 2 with no additional equilibrium assumptions?

We have made changes to the earlier parts of the manuscript to motivate and justify the equilibrium assumptions. This is so particularly in the sections describing the model set-up, and by clarifying that sickness is a property of the resources being traded rather than the players themselves. Additionally, at the end of the model description there is some justification for the model set-up, we have added the following to this:

In the case of an endemic disease that is maintained by a combination of within-group spread and trading, this "conservation law" is a reasonable approximation of a scenario with constant prevalence.

We thank the reviewer for their thorough feedback and careful reading of our manuscript.

Reviewer #2:

This work tries to pose a quite new type of ‘vaccination game’. When one calls a vaccination game, it implies models based on mathematical epidemiology; such as SIS, SIR and so forth, to depict disease spreading process, combined with the framework of evolutionary game theory to reproduce an individual decision whether committing a vaccination or not depending on the balance between vaccination cost and potential risk to be infected unless vaccinated (aiming ‘free-ride’ to the so-called herd immunity). As well-known, there have been quite a few vaccination game (VG) models in the sense of conventional VG abovementioned. What the authors establishing here has a entirely different scope. In this sense, it is said quite new. I would say it is an admirable trial, although I see it quite specific and less universal vis-à-vis usual VG models. The game framework is depicted twofold relations of agents (players) that are connected underling network that may be time-evolving (taking place refreshments of a random link like dynamic small world network) or time-frozen. The twofold relations are what they are called; ‘exchange of money’ (more fairly to be said ‘resource’) and ‘exchange of sickness’.

We have replaced the word "money" with "tokens" and used resource in a slightly different context as it is helpful to clarify the description of the game.

A game proceeds with each of agent, obeying to his/ her strategy exchanging resource. On the process of such exchange, disease spreads. Quite interestingly (or strangely) to me, when an agent successfully passes a unit of sickness to a contact person, the contact person may be infected. That’s fine, since it implies a transmittance process between I and S states. Yet, the focal agent who gives such unwilling gift to the other person is able to successfully lose a unit of sickness. I’m not so sure what this idea implying in relation with what’s happening in real world.

The following passages have been added to the introduction:

Unlike the more catastrophic epidemic threats, endemic disease is perceived by some farmers as a tolerable loss to farm productivity and animal welfare; a hard-to-avoid management problem that comes and goes in varying amounts as disease carrying livestock are continually imported in and exported away to keep the enterprise commercially functional. The novelty of this situation, when framed as a problem of the “vaccination game” type, is the motivation behind this paper.

And

The game assumes that trading is an unavoidable component of a livestock production sector, and so of the livelihood of individual farmers. Each individual must make a purchase every turn. They cannot protect themselves through isolation or biosecurity (as in typical vaccination games) and instead must decide which other individuals they wish to be exposed to.

To make this MS more impressive to the audience who is interested in VG models, I think the authors should more carefully and deliberately highlight how their model is new and different from the usual VG models by citing some those representative (and quite recent ones), for instance;

Wang, Bauch, Bhattacharyya, d’Onofrio, Manfredi, Perc, et al., Statistical physics of vaccination, Physics Report 664, 1-133, 2016.

Arefin et al.; Interplay between cost and effectiveness in influenza vaccination uptake: vaccination game approach, Proceedings of the Royal Society A 475, 20190608, 2019.

Kabir et al.; Modelling and analysing the coexistence of dual dilemmas in the proactive vaccination game and retroactive treatment game in epidemic viral dynamics, Proceedings of the Royal Society A 475, 20190484, 2019.

<https://doi.org/10.1098/rspa.2019.0484>

Kabir et al.; Behavioral incentives in a vaccination-dilemma setting with optional treatment, Physical Review E 100, 062402, 2019.

Evolutionary Games with Sociophysics: Analysis of Traffic Flow and Epidemics, Springer, 2019.1; ISBN: 978-981-13-2769-8

Fundamentals of Evolutionary Game Theory and its Applications, Springer, 2015.10; ISBN: 978-4-431-54961-1

The following has been added including references to the suggested papers:

This dynamic has motivated a number of mathematical models, some incorporating the effects of financial cost, social influence, and network structure [12-14]., and commonly using methods from statistical physics [15]. This class of models has recently expanded beyond simple vaccination to encompass a nuanced range of public health concerns [16,17].

We have also added this to the following paragraph:

This is particularly true when we consider endemic livestock diseases. Unlike the more catastrophic epidemic threats, endemic disease is perceived by some farmers as a tolerable loss to farm productivity and animal welfare; a hard-to-avoid management problem that comes and goes in varying amounts as disease carrying livestock are continually imported in and exported away to keep the enterprise commercially functional [20]. The novelty of this situation, when framed as a problem of the "vaccination game" type, is the motivation behind this paper.

We thank the reviewer for their thorough feedback and careful reading of our manuscript.

Board member:

Please address reviewers' points. It would be beneficial to expand on motivation and highlight again at the beginning of Model section that you are dealing with endemic diseases as opposed to epidemic (as mentioning epidemic models in the intro seem to nudge toward a wrong perception of your setting). Further clarification of the initialisation of the game and different parameters' ranges is needed.

We thank the board member for their insightful suggestions. We have taken every care to address the reviewers concerns and hope that our work now meets the standard of the journal. We have made it clear in the Introduction that our focus is on an endemic disease situation, as you suggest.

Dear Editor of Proceedings A,

We are very grateful for the attention given to our manuscript. We revised the manuscript following the remaining suggestions of reviewer #1 and believe the manuscript meets the requirements of the editor. Please see our detailed replies to these comments below.

Kind regards,

Ewan Colman,
Nick Hanley,
Rowland Kao

Note: The reviewers' comments are indented.

Reviewer #1:

The authors have addressed many of comments from the previous version. However, some of their responses do not address the comments and appear to side step around the issue (or are mathematically incorrect).

We are grateful to the reviewer the care that they have put into reviewing our manuscript. It was not our intention to side-step any of the issues raised and we hope that we can be forgiven for accidentally overlooking these problems in the first revision.

1) The authors state:

Since R_t cannot diverge as t increases ($R_t < N$ for all t), we assume that there exists R such as $R_t \rightarrow R$ for large values of t .

There is still no justification for this assumption. Even if R_t cannot diverge, it may asymptotically approach periodic orbits or even exhibit chaotic dynamics. Why do they assume that there is a globally asymptotically stable equilibrium point?

This is indeed true if R_t represents the value of an individual run of the model (i.e. one of the lines in Figure 1), however R_t was intended to represent the expected value of mean of the distribution of the number of rewired links over many runs of the model. We have edited this section so that R_t is now introduced as the value in one model run and have used the angular bracket notation to represent the mean with the following added...

We use $\langle R_t \rangle$ to denote the mean value of R_t we would expect to find over a large number of realisations of the model.

2) In their response, the authors state:

The range of N is now given in Table 1. There is nothing about equation (3.3) that requires N to be very large.

I frankly don't understand the authors' response here. Equation 3.3 is derived using a Taylor approximation for r close to 0 (or N approaching infinity), as is stated in the manuscript. I

repeat my previous comment. The authors should provide a better derivation and justification for (3.3) perhaps in an appendix? How large does N need to be for this to hold?

In the previous round of revisions we had mistakenly thought that this comment was directed at Eq.(3.2). We thank the reviewer for their persistence in making sure this is correct.

The explanation of this derivation has now been expanded. The justification for making the approximation is described in the paper after equation (3.2).

3) In their response, the authors state,

Yes. The absence of token exchange is a mechanism to incentivise the choice of low demand suppliers in a way that mimics real market dynamics. This is now clearly stated in terms of resources: Each player has some resource that they trade with other players for tokens, each unit of resource may be sick or healthy.

I suggest they also add a statement to the manuscript describing why they model no exchange of token as written above.

The second to last paragraph of the first part of the Model section now includes these details...

The exchange of tokens incorporates a simple mechanism of supply and demand into the model; losses (in terms of tokens) are incurred when a player encounters other players wanting to buy from the same source as them. The rule that token exchange does not occur when the seller is uniquely chosen by the buyer creates an incentive to choose from players who are unlikely to be chosen by others.

Reviewer #2:

The revised version seems clear and appropriate for publication..

We thank the reviewer for their thorough feedback and careful reading of our manuscript.

Board member:

We thank the authors for submitting a revision. As can be seen from the reviews, there are still some issues remaining for authors to address. Given the importance of the topic during the time of the pandemic, we decided to give authors another opportunity to respond to reviews. Please check the reviews for detailed comments.

We thank the board member for the attention they have given to the manuscript. We have taken every care to address the reviewer's concerns and hope that our work now meets the standard of the journal.